A comparison of blood gases, biochemistry, and hematology to ecomorphology in a health assessment of pinfish (Lagodon rhomboides)

Collins Sara 1
Dornburg Alex 2
Flores Joseph M. 2
Dombrowski Daniel S. 3
Lewbart Gregory A. greg_lewbart@ncsu.edu 4
1 College of Veterinary Medicine, University of Georgia , Athens , GA , United States
2 Research and Collections, North Carolina Museum of Natural Sciences , Raleigh , NC , United States
3 Veterinary Services Unit, North Carolina Museum of Natural Sciences , Raleigh , NC , United States
4 Clinical Sciences, North Carolina State University College of Veterinary Medicine , Raleigh , NC , United States
Zhao Min
Electronic publication date: 2016 Aug 9
Publication date: 2016
Volume: 4
Electronic Location ID: e2262
Received 2016 Feb 19; Accepted 2016 Jun 26
Copyright: ©2016 Collins et al.
Copyright year: 2016
Copyright holder: Collins et al.
License: This is an open access article distributed under the terms of the Creative Commons Attribution License, which permits unrestricted use, distribution, reproduction and adaptation in any medium and for any purpose provided that it is properly attributed. For attribution, the original author(s), title, publication source (PeerJ) and either DOI or URL of the article must be cited.
License URL: https://creativecommons.org/licenses/by/4.0/

Keywords: Sparidae, Hematology, Geometric morphometrics, Plasma biochemistry, Feeding ecology

Funding: NC State College of Veterinary Medicine This work was supported in part by the Robert J. Koller Aquatic Animal Medicine endowment at the NC State College of Veterinary Medicine. The funders had no role in study design, data collection and analysis, decision to publish, or preparation of the manuscript.

==============================
Despite the promise of hematological parameters and blood chemistry in monitoring the health of marine fishes, baseline data is often lacking for small fishes that comprise central roles in marine food webs. This study establishes blood chemistry and hematological baseline parameters for the pinfish Lagodon rhomboides, a small marine teleost that is among the most dominant members of near-shore estuarine communities of the Atlantic Ocean and Gulf of Mexico. Given their prominence, pinfishes are an ideal candidate species to use as a model for monitoring changes across a wide range of near-shore marine communities. However, pinfishes exhibit substantial morphological differences associated with a preference for feeding in primarily sea-grass or sand dominated habitats, suggesting that differences in the foraging ecology of individuals could confound health assessments. Here we collect baseline data on the blood physiology of pinfish while assessing the relationship between blood parameters and measured aspects of feeding morphology using data collected from 37 individual fish. Our findings provide new baseline health data for this important near shore fish species and find no evidence for a strong linkage between blood physiology and either sex or measured aspects of feeding morphology. Comparing our hematological and biochemical data to published results from other marine teleost species suggests that analyses of trends in blood value variation correlated with major evolutionary transitions in ecology will shed new light on the physiological changes that underlie the successful diversification of fishes.

Introduction

Blood chemistry and hematological parameters have become an increasingly important component of monitoring the health of wild populations (Brenner et al., 2002; Anderson et al., 2010; Lewbart et al., 2014). Blood based health assessments are ideally suited for marine environments, where safety limitations in scientific diving (Dardeau & McDonald, 2007) and differences in the efficiency of survey methods across taxa (Willis, Miller & Babcock, 2000), have stymied the ability to monitor changes in the physiological health of difficult to observe organisms. With current or potential future changes in marine species population dynamics increasingly linked to a host of contemporary threats including acidification (Orr et al., 2005; Hoegh-Guldberg et al., 2007; Fabry et al., 2008), warming (Stachowicz et al., 2002; Schiel, Steinbeck & Foster, 2004; Near et al., 2012a), or invasion of non native species (Thatje et al., 2005; Albins & Hixon, 2013), the ability to rapidly survey physiological changes in a minimally invasive manner holds promise for marine conservation efforts. In particular, collecting data on species that form critical links in marine food webs provides the baseline infrastructure for assessing, or forecasting, changes in the wake of environmental disaster, unexplained morbidity/mortality events, or disease outbreaks (Seaward, 1994; Lohner et al., 2001; Harvell et al., 2002).

In the Western Atlantic, pinfish (Lagodon rhomboides) are among the most dominant members of near-shore estuarine communities (Reid Jr, 1954; Hansen, 1969; Stoner, 1980), and are ubiquitous across a variety of habitats from New England to Florida as well as the northern Gulf of Mexico to the Yucatan Peninsula (Orth & Heck, 1980; Stoner & Livingston, 1984; Bonilla-Gómez et al., 2011). With seasonal abundances capable of altering the composition of estuarine epifaunal seagrass communities (Stoner, 1982), these fish form the prey base for many larger fishes at higher trophic levels (Schmidt, 1986; Bethea et al., 2006). This abundance has made pinfish a popular live bait used by recreational and commercial fisherman (Muncy, 1984; Adams et al., 1998) with high market demands driving the development of these fish as a potential new aquaculture species (DiMaggio et al., 2010; Ohs et al., 2010; DiMaggio, Broach & Ohs, 2013). Given their increasing economic importance, central role in near-shore marine food webs, and high abundances, pinfish are ideal candidates for comprehensive and minimally invasive health assessments from which to monitor changes in marine communities.

Although pinfish exhibit many of the hallmarks of a model species, there is also evidence for substantial differences in morphology and feeding ecology of these fishes at fine-spatial scales (Ruehl, Shervette & Dewitt, 2011). In particular, preferential colonization and settlement in sandy versus eel-grass habitats have been suggested to be a primary axis of selection in pinfish, driving pronounced changes in jaw morphology between geographically proximate sites (Ruehl, Shervette & Dewitt, 2011). These changes in jaw morphology occur across all size classes and reflect shifts in diet between seagrass and open sand-flat communities (Luczkovich, 1988; Levin, Petrik & Malone, 1997; Ruehl, Shervette & Dewitt, 2011). Since changes in feeding ecology have been demonstrated to affect blood parameters in other species such as seals (Thompson et al., 1997), this makes accounting for habitat specificity a potentially necessary step towards developing baseline health models for this species in any site comprised of heterogeneous habitat types.

Here we conducted a health assessment of pinfishes, measuring blood gas, biochemistry, and hematology values in an area composed of a mixture of sandy and eel grass habitats from 37 individual fishes. Our results were consistent with known blood parameters while greatly expanding upon measured hematological and biochemical data on the species. We further integrated our blood analyses with a combination of linear and geometric morphometric analyses to test for possible relationships between ecomorphology and blood physiology. To the best of our knowledge, our study is the first to test for potential correlations between ecomorphology and blood physiology within a species of marine fish and provides a new perspective on baseline data for this ecologically important species.

Materials and Methods

Ethics statement

This study was conducted at the North Carolina State University Center for Marine Sciences and Technology (CMAST) in Morehead City, North Carolina and approved by the IACUC ethics and animal handling protocol. All handling and sampling procedures were consistent with standard vertebrate protocols and veterinary practices.

Pinfish capture

Pinfish were captured in Bogue Sound (34.721734°N, −76.759587°W) via lightweight angling gear. We aimed to capture 40 individual fish from this population for baseline estimates. This sampling strategy balanced budget constraints with our ability to capture the variance of quantified parameters and is similar to other health assessments of wild fish (Fazio et al., 2013). Thirty-seven fish were successfully landed with fight times lasting an average of less than 15 s. Once captured, the fish were placed in a 20 L plastic bucket with freshly aerated seawater from the collection site. The fish were immediately transported to the laboratory adjacent to our sampling site, with a transport time of less than 3 min. Once in the laboratory, fish were maintained in aerated seawater from the sample site in individual 20 L buckets until they were sedated with MS-222. Pinfish were left to acclimatize for 5 min after transport and before induction.

Blood sample collection and handling

All fish were sedated with buffered MS-222 (100 mg/L) and blood samples were collected intravenously from the coccygeal hemal arch into pre-coated, heparinized U-100 insulin syringes with BD Ultra-Fine™ needles in the 1/2 mL (50 unit) size. An average of 0.23 mL was obtained from each fish (standard deviation = 0.05 mL, median = 0.22 mL; see Data S1). The blood was then loaded into the CG-8+ iSTAT cartridges (Abaxis Corporation, Union City, California, USA) and lactate analyzer (Nova Biomedical Corporation, Waltham, Massachusetts, USA) within 5 min of sample collection. All blood values were obtained within 5 min of sample collection.

Blood gas and biochemistry parameters

The biochemistry, blood gas, and electrolyte results were obtained using an iSTAT Portable Clinical Analyzer with CG8+ cartridges. The iSTAT is a portable, handheld, battery-operated electronic device with the ability to measure a wide variety of chemistry, blood gas, and basic hematology parameters with only a few drops (0.095 mL) of whole, non-coagulated blood. The following parameters were measured and recorded: pH, lactate, pO2, pCO2, HCO32, Hct, Hb, Na, K, iCa, and glucose. The iSTAT device analyzed the blood at 37 °C. Using the equations provided by Mandelman & Skomal (2009), we manually calculated an independent set of corrections for pH based on the water temperature (T) at the time of sampling (27 °C): (1) pHTC=pHM−0.011T−37.

Corrections for calcium were additionally made using the following equation from Mandelman & Skomal (2009): (2) iCaTC=iCaM1+−0.53pHTC−pHM

For Eqs. (1) and (2), values measured from the iSTAT are denoted by an “M” subscript. Those that were manually temperature corrected by using the equations below are denoted by “TC” subscript. Following the guidelines by Friedrichs et al. (2012), reference intervals were computed using the robust method in the referenceInterval package in R, as Shapiro–Wilks tests (Shapiro & Wilk, 1965) suggested blood values to not follow the expectations of a Gaussian distribution. This distribution-independent method of calculating a reference interval is preferred for studies of wild populations with low samples sizes (Friedrichs et al., 2012). Prior to reference interval calculation, outliers were identified and removed using the method of Horn et al. (2001). Confidence intervals of the upper and lower bounds of the reference intervals were calculated using 5000 bootstrap replicates.

Hematology

Hematocrit was determined using high-speed centrifugation of blood-filled hematocrit tubes with a Zipocrit Hematocrit Centrifuge (ThermoFisher Scientific, Philadelphia, PA). All white blood cell (WBC) count estimates were performed by the same technician, at a location on the slide where the cells were one layer thick, adjacent to one another (membranes touching), evenly distributed, and showed no signs of morphological changes (Newman, Piatt & White, 1997). White blood cell estimates were made by using a 100X objective lens with immersion oil, counting the number of white blood cells in 10 fields, calculating the average, and then multiplying the number of cells by 2,000 (Gaunt et al., 1995). Using a 100X objective lens with immersion oil, differential white blood cell counts were performed by examining 100 white blood cells on a peripheral smear stained with Wright-Giemsa stain and counting the number of lymphocytes, neutrophils, monocytes, and eosinophils (Fig. 1). The absolute cell count for each type of cell was calculated by multiplying the percentage of the type of cell by the overall WBC estimate (Newman, Piatt & White, 1997).

Figure 1 Blood smear illustrating the different cell types (Wright-Giemsa stain, 1000X).

Specimen digitization and dissection

Following blood collection, fish were euthanized using an additional dose of MS-222, at a concentration of >250 mg/L. The fish were left in this solution for at least 10 min following cessation of opercular movement. Following euthanasia all specimens were photographed facing left, using a 12 megapixel camera. For each specimen two sets of photographs were taken. Specimens were first photographed with their mouths closed, followed by a second round of photographs taken with mouths opened to their maximum level of jaw protrusion. Following digitization, specimens were dissected and reproductive organs were examined to accurately assign sex to each individual to test for the potential of sexual dimorphism in trait data. All specimens were subsequently deposited in the Ichthyological Collection of the North Carolina Museum of Natural Sciences (NCSM 81424).

Quantifying body shape

Body shape was quantified from the digitized images using landmark-based geometric morphometric methods (Bookstein, 1997; Adams, Rohlf & Slice, 2004; Zelditch, Swiderski & Sheets, 2012) in the TpsDIG2 software package (Rohlf, 2005). To quantify body shape, 27 homologous landmarks used in other fish morphometric studies (Dornburg et al., 2011; Frederich et al., 2012) were used to capture body shape variation (Fig. 2A). Briefly these are: (1) posteroventral corner of the maxilla; (2) anteroventral tip of the premaxilla; (3) anterodorsal point of mouth where fleshy lip meets scales; (4) most anterior point of eye; (5) most dorsal point of eye; (6) most posterior point of eye; (7) most ventral point of eye; (8) center of eye; (9) anterior point of first dorsal spine insertion; (10) dorsal fin origin; (11) posterior point where dorsal fin sheath joins fin rays; (12) dorsal fin insertion; (13) dorsal inflection of caudal peduncle; (14) dorsal caudal-fin ray insertion; (15) ventral caudal-fin ray insertion; (16) ventral inflection of caudal peduncle; (17) anal fin insertion; (18) anal fin origin; (19) posterior point of pelvic fin insertion; (20) anterior point of pelvic fin insertion; (21) ventral insertion of the operculum; (22) posterior ventral point where fleshy lower lip meets scales; (23) anterodorsal point of lower jaw; (24) dorsal point where pectoral fin base joins body; (25) dorsal point of pectoral fin ray insertion; (26) ventral point of pectoral fin ray insertion; (27) ventral point where pectoral fin base joins body. To better capture the curves of the body between landmarks, five sliding semi-landmarks were placed as follows: (1) at the midpoint between landmarks 3 and 9; (2) at the midpoint of the dorsal fin, placed along the body; (3) at the midpoint between the dorsal and ventral caudal fin ray insertions, placed along the fin ray insertion margin; (4) at midpoint of the anal fin, placed along the body; and (5) at the midpoint between landmarks 21 and 22.

Figure 2 (A) Placement of homologous landmarks (light circles) and sliding semi-landmarks (dark circles). (B) Linear and angular jaw measurements taken from each individual.

Linear measurements and angles of protrusion

Linear measurements were taken on each digitized specimen using the ImageJ software package (Abràmoff, Magalhães & Ram, 2004). For each digitized specimen, the images of corresponding to maximum jaw opening were used to measure standard length, maximum protrusion of the premaxilla, length of the mandible, and maximum gape size. Since pinfish ecomorphs are divided into groups that reflect changes in jaw morphology correlated with changes between bottom and water column feeding (Ruehl, Shervette & Dewitt, 2011), we additionally quantified three angles to assess the mouth position of each specimen during feeding (Fig. 2B). The downward angle of the upper jaw was measured drawing a line along the ventral margin of the head just dorsal to the maxilla, down the dorsal margin of the premaxilla, to a termination at the most anterior point of the premaxilla (Fig. 2B). Similarly, the angle of the lower jaw was measured by drawing a line along the ventral margin of each specimen from the operculum to the angular, with a connecting segment connecting to the most anterior point of the dentary along its ventral margin (Fig. 2B). Changes in the midpoint of the jaw position were quantified by drawing a line along the ventral margin of the head just dorsal to the maxilla from it’s most anterior to its most ventral point, with a connecting segment to the most ventral point of the premaxilla (Fig. 2B).

Statistical analysis

To test for sexual dimorphism in our three trait datasets we used a combination of principle component analysis (PCA) and multivariate analysis of variance (MANOVA). For the body shape data, a Procrustes fit was first used to remove variation due to scaling, rotation, and translation (Rohlf & Slice, 1990; Zelditch, Swiderski & Sheets, 2012) in the body shape data. Procrustes coordinates were subjected to a PCA implanted in the R package geomorph (Adams & Otárola-Castillo, 2013). Although relative warps analyses are often alternatively applied to coordinate data (Bastir & Rosas, 2006; Sidlauskas, 2008; Dornburg et al., 2011), a PCA is equivalent to a relative warps analysis with an alpha set to 0 (Rohlf, 1993; Birch, 1997).

For both the jaw dataset comprising linear and angle measurements, and the hematological dataset, data were log transformed and first regressed against log body size to account for the possibility of allometry. To account for the possibility of different allometric trends between males and females, regressions were conducted with the trait data separated by sex. A PCA was then conducted on the residuals of the regressions for each dataset. Morpho- and hematospaces were generated for each PCA by plotting the orthogonal eigenvectors that correspond with the major axes of shape variation, with convex hulls of the male and female data plotted to visualize the degree of overlap. For each class of data, a MANOVA was used to compare the PC axes that cumulatively summed to 95% of the variance between male and female Lagodon rhomboides.

We built four models that compared the effect of (1) Feeding morphology; (2) Feeding morphology with size as a covariate; (3) Body shape (PC1); and (4) a null intercept-only model on each blood value trait. The fits of all models were simultaneously compared using general linear models (Nelder & Wedderburn, 1972) in conjunction with an information theoretic framework based on Akaike’s Information Criterion (Akaike, 1973) corrected for small sample size (Burnham & Anderson, 2002). Sample size corrected AICc weights (wi) and coefficients were estimated using Akaike weight based model averaging across all models. For feeding ecology, lower jaw angle was used as an alternative to qualitative assessment of how terminal versus inferior the mouth position was. All analyses were conducted in R using the libraries bbmle (Bolker, 2010) and MuMIn (Barton & Barton, 2015).

Results

Hematological values

Tables 1 and 2 display the biochemistry, blood gas, and hematology results for the 37 pinfish analyzed. We collected similar numbers of males and females (Table 1), with the values of all parameters overlapping between sexes (Fig. S1). In a few samples the iSTAT blood analyzer was unable to calculate the values for some of the parameters resulting in slightly smaller n (Table 1). Also, for certain parameters, the iSTAT indicated values that exceeded the maximum detectable (Table 1). For the purpose of calculating a mean group value the maximum recordable value was used. While this yields a lower estimate of the mean than actually exists within the data, the low numbers of samples above the maximum recordable level (Table 1) likely produce only a minor skew to the estimated distribution of variation as measurable values remained well between the first and third quartiles (Table 1).

Table 1 Descriptive statistics of the blood gas and blood biochemical values collected.

Analyte	N (m/f)	Mean, SD	Quartile (25%, 50%, 75%)	Min (OOR)	Max (OOR)	Reference interval	90% CI of lower RI	90% CI of upper RI	
Na (mmol/L)	37 (14/17)	168.84, 7.04	166, 169, 179	156	>180 (6)	154.0, 187.3	149.4, 157.3	183.3, 191.0	
K (mmol/L)	31 (14/12)	4.7, 1.4	3.9, 4.6, 7.4	3.1	>9.0 (6)	0.26, 9.01	0, 1.28	7.27, 10.6	
iCa (TC) (mmol/L)	37 (14/17)	1.56, 0.18	1.44, 1.59, 1.66	1.23	>2.5 (1)	1.15, 1.96	1.04, 1.23	1.89, 2.07	
Glucose (mg/dl)	37 (14/17)	169.7, 101.43	84, 155, 217	45	419	0, 373.9	0, 0	307, 444	
Hct (%)	31 (12/14)	34.39, 9.54	30.5, 34, 37.5	17	58	25.0, 43.6	22.2, 27.6	40.7, 46.8	
Hb (g/L)	31 (12/14)	11.7, 3.24	10.35, 11.6, 12.75	5.8	19.7	8.59, 14.8	7.63, 9.50	13.9, 15.9	
pH	37 (14/17)	7, 0.28	6.85, 7.14, 7.32	<6.5 (1)	7.505	6.47, 7.68	6.30, 6.62	7.52, 7.86	
Beecf	36 (14/16)	−20.78, 5.56	−29.25, −23.5, −19.5	<−30 (9)	−10	−37.5, −10.2	−13.7, −5.1	−40.0, −31.4	
HCO3 (mmol/L)	36 (14/16)	7.52, 3.07	4.92, 7.2, 9.32	3.3	13.8	0.52, 13.4	−0.97, 2.36	11.6, 15.5	
TCO2 (mmHg)	36 (14/16)	8.94, 2.84	6, 8, 11	<5 (6)	14	1.97, 14.0	0.89, 3.75	12.1, 16.0	
sO2% (mmHg)	29 (10/13)	19.79, 15.14	6, 16, 32	3	49	0, 44.8	0, 0	34.2, 56.5	
Lactate (mmol/L)	37 (14/17)	9.93, 4.56	6.4, 9.9, 13.9	2	17.6	2.48, 17.1	0.63, 4.29	15.2, 19.3	
PCV	34 (13/16)	53.18, 12.06	45.25, 50.5, 60	30	80	26.8, 76.2	20.4, 32.7	69.2, 85.2	
Notes.

N number of samples

m males

f females

SD standard deviation

Min minimum value

Max maximum value

OOR number of samples outside of the recordable range of our instruments

RI reference interval

CI confidence interval

Reference Intervals were truncated with a lower bound of zero to maintain biological realism in non-negative blood parameters.

Table 2 Descriptive statistics of the manually analyzed hematology parameters collected.

Cell type	N (m/f)	Mean (SD)	Quartile (25%, 50%, 75%)	Min	Max	Reference interval	90% CI of lower RI	90% CI of upper RI	
Total WBC/ul	30 (14/13)	25,788 (14,395.68)	16,200, 20,000, 32,400	6,800	73,200	0, 54,141	0, 1647	44,684, 68,230	
Lymphocyte (%)	30 (14/13)	80.5 (15.96)	74.25, 87, 91	39	96	52.1, 123.0	39.5, 63.2	113.6, 135	
Neutrophil (%)	30 (14/13)	17.1 (14.86)	7, 11, 22.5	2	54	0, 44.0	0, 0	34.0, 55.5	
Monocyte (%)	17 (7/8)	3.8 (2.51)	2, 3, 5	0	11	0, 8.72	0, −5.78	6.23, 11.7	
Notes.

N number of samples

m males

f females

SD standard deviation

Min minimum value

Max maximum value

RI reference interval

CI confidence interval

Reference Intervals were truncated with a lower bound of zero to maintain biological realism in non-negative blood parameters.

Statistical analysis of sexual dimorphism

Pinfish measured ranged in size between 93.8 and 168 mm (median = 137.7 mm; 1st quartile = 130.5 mm, 3rd quartile = 150.6 mm) and in weight between 11 and 148 g (median = 80 g; 1st quartile = 58 g, 3rd quartile = 93 g). Jaws measured displayed a 20°and 30°range of angles for the upper and lower jaws respectively, with no predicted relationship between jaw angle and size (Table S1), corresponding with previous work demonstrating substantial differences in jaw orientation in relation to habitat (not allometry) (Ruehl, Shervette & Dewitt, 2011).

Table 3 Results of GLM fitting to test predictive power of morphological data on hematological parameters.

Set	Response variable	Predictor variable	df	ΔAICc	wi	Model-averaged coefficients (95% CI)	
1	Sodium	Intercept only	2	0.0	0.43	−3.09e–4(−0.05, 0.05)	
	Sodium	Ecomorph (lower jaw angle)	3	0.8	0.31	0.09(−1.52, 2.07)	
	Sodium	Ecomorph (lower jaw angle) * size	5	5.5	0.24	0.115(−0.41, 1.37)	
	Sodium	Body shape (PC1)	3	10.6	0.02	−0.008(−1.55, 0.92)	
2	Potassium	Intercept only	2	0.0	0.56	2.23e–3(−0.46, 0.47)	
	Potassium	Ecomorph (lower jaw angle)	3	1.6	0.20	−0.476(−19.62, 16.00)	
	Potassium	Body shape (PC1)	3	2.4	0.09	−0.321(−10.67, 6.95)	
	Potassium	Ecomorph (lower jaw angle) * size	5	7.4	0.01	0.028(−13.11, 17.26)	
3	Calcium	Intercept only	2	0.0	0.54	4.46e–4(−0.13, 0.13)	
	Calcium	Body shape (PC1)	3	1.3	0.28	0.431(−1.49, 4.56)	
	Calcium	Ecomorph (lower jaw angle)	3	2.4	0.16	−0.011(−5.36, 5.23)	
	Calcium	Ecomorph (lower jaw angle) * size	5	7.8	0.01	−0.001(−4.45, 4.21)	
4	Glucose	Intercept only	2	0.0	0.56	4.90e–4(−0.71, 0.71)	
	Glucose	Ecomorph (lower jaw angle)	3	1.5	0.26	−0.544(−25.11, 21.19)	
	Glucose	Body shape (PC1)	3	2.4	0.17	−0.075(−14.09, 13.18)	
	Glucose	Ecomorph (lower jaw angle) * size	5	7.0	0.01	0.01(−18.36, 19.49)	
5	Hematocrit	Ecomorph (lower jaw angle)	3	0.0	0.38	0.046(−0.65, 0.74)	
	Hematocrit	Intercept only	2	0.3	0.32	−2.457(−33.26, 21.99)	
	Hematocrit	Body shape (PC1)	3	0.9	0.24	0.993(−2.09, 10.38)	
	Hematocrit	Ecomorph (lower jaw angle) * size	5	3.9	0.06	0.356(−3.73, 16.92)	
6	Hemoglobin	Ecomorph (lower jaw angle)	3	0.0	0.36	0.044(−0.64, 0.73)	
	Hemoglobin	Intercept only	2	0.2	0.34	−2.335(−33.01, 21.92)	
	Hemoglobin	Body shape (PC1)	3	0.8	0.25	0.993(−2.14, 10.32)	
	Hemoglobin	Ecomorph (lower jaw angle) * size	5	3.9	0.05	0.339(−3.77, 16.9)	
7	pH	Intercept only	2	0.0	0.47	2.33e–4(−0.05, 0.05)	
	pH	Body shape (PC1)	3	0.8	0.31	−0.188(−1.54, 0.36)	
	pH	Ecomorph (lower jaw angle)	3	1.7	0.20	−0.012(−2.23, 2.12)	
	pH	Ecomorph (lower jaw angle) * size	5	6.6	0.02	7.72e–3(−0.89, 1.79)	
8	PCO2	Intercept only	2	0.0	0.55	7.49e–4(−0.38, 0.38)	
	PCO2	Body shape (PC1)	3	1.7	0.24	0.795(−4.73, 11.33)	
	PCO2	Ecomorph (lower jaw angle)	3	2.0	0.20	−0.145(−14.67, 13.33)	
	PCO2	Ecomorph (lower jaw angle) * size	5	7.4	0.01	3.69e–3(−11.73, 11.18)	
9	PO2	Body shape (PC1)	3	0.0	0.38	−6.51e–3(−0.75, 0.74)	
	PO2	Intercept only	2	0.2	0.33	−4.14(−24.82, 2.88)	
	PO2	Ecomorph (lower jaw angle)	3	0.7	0.27	0.906(−21.21, 27.54)	
	PO2	Ecomorph (lower jaw angle) * size	5	6.1	0.02	−0.024(−21.08, 18.33)	
10	BEEcf	Intercept only	2	0.0	0.55	4.45e–4(−0.35, 0.36)	
	BEEcf	Ecomorph (lower jaw angle)	3	1.5	0.25	0.187(−11.15, 12.52)	
	BEEcf	Body shape (PC1)	3	2.3	0.18	−0.218(−8.12, 5.62)	
	BEEcf	Ecomorph (lower jaw angle) * size	5	7.0	0.02	0.012(−8.86, 10.25)	
11	Bicarbonate	Intercept only	2	0.0	0.61	8.62e–4(−0.41, 0.41)	
	Bicarbonate	Ecomorph (lower jaw angle)	3	2.3	0.20	−0.178(−16.35, 14.66)	
	Bicarbonate	Body shape (PC1)	3	2.4	0.18	−0.013(−8.92, 8.78)	
	Bicarbonate	Ecomorph (lower jaw angle) * size	5	7.7	0.01	0.015(−11.3, 13.54)	
12	Total CO2	Intercept only	2	0.0	0.57	9.82e–4(−0.36, 0.36)	
	Total CO2	Ecomorph (lower jaw angle)	3	1.8	0.23	−0.254(−13.7, 11.64)	
	Total CO2	Body shape (PC1)	3	2.3	0.18	0.208(−6.15, 8.47)	
	Total CO2	Ecomorph (lower jaw angle) * size	5	7.2	0.02	0.012(−9.42, 11)	
13	Saturated 02%	Intercept only	2	0.0	0.60	−0.025(−1.24, 1.19)	
	Saturated 02%	Ecomorph (lower jaw angle)	3	2.4	0.182	2.132(−61.75, 81.97)	
	Saturated 02%	Body shape (PC1)	3	2.4	0.181	−0.726(−24.42, 16.41)	
	Saturated 02%	Ecomorph (lower jaw angle) * size	5	6.1	0.03	−0.492(−42.6, 8.01)	
14	Lactate	Body shape (PC1)	3	0.0	0.50	−2.06e–3(−0.6, 0.59)	
	Lactate	Intercept only	2	0.8	0.34	4.801(−1.41, 20.57)	
	Lactate	Ecomorph (lower jaw angle)	3	2.5	0.14	0.439(−31.45, 37.02)	
	Lactate	Ecomorph (lower jaw angle) * size	5	6.8	0.02	−0.132(−23.74, 7.87)	
15	Packed cell volume	Intercept only	2	0.0	0.59	−4.72e–4(−0.24, 0.24)	
	Packed cell volume	Ecomorph (lower jaw angle)	3	2.0	0.22	−0.074(−8.82, 8.18)	
	Packed cell volume	Body shape (PC1)	3	2.4	0.18	−0.066(−5.31, 4.56)	
	Packed cell volume	Ecomorph (lower jaw angle) * size	5	7.5	0.01	−6.03e–3(−7.43, 6.56)	
16	WBC	Intercept only	2	0.0	0.47	0.005(−0.56, 0.57)	
	WBC	Body shape (PC1)	3	0.9	0.29	2.06(−4.61, 18.7)	
	WBC	Ecomorph (lower jaw angle)	3	1.4	0.23	−0.29(−17.9, 15.51)	
	WBC	Ecomorph (lower jaw angle) * size	5	7.3	0.01	−0.01(−16.25, 13.55)	
Notes.

Df degrees of freedom

ΔAIC difference in AIC score between the model and the best-fitting model

wi AIC weight

df degrees of freedom

WBC white blood cell count

Principle component analysis of the body shape data provided no evidence of sexual dimorphism. Principle component axes one through five respectively captured 24, 17, 7, 6, and 5 percent of the cumulative variance with the first axis of body shape change describing an elongation of the caudal peduncle coupled with an anterodorsal widening of the cranium (Fig. 3A). The second PC axis described changes in the anterior slope of the cranium coupled with an anteroposterior elongation of the body and cranium (Fig. 3A). There was no clear separation of male and female body shapes in morphospace with a MANOVA on the PC scores finding no significant effect of sex on body shape (Pillai’s trace = 0.485, F = 1.79, p = 0.082).

Figure 3 Visualizations of the first two principle component axes for changes in (A) body shape quantified by the geometric morphometric analyses; (B) jaw morphology; and (C) hematological data. Outlines correspond to the convex hull of the morphospace for females or males.

Similarly, there was no clear separation of male and females in either the jaw morphospace (Fig. 3B) or the hematospace (Fig. 3C). In the jaw PCA, the first three PC axes captured 87 percent of the variance (PC1 = 56%; PC2= 19%; PC3 = 12%) while a PCA of the hematological data yielded three axes that together described 85% of the cumulative variance (PC1 = 48%; PC2 = 26%; PC3= 11%). A MANOVA on the PC scores mirrored the qualitative results of the morphospaces and provided no support for a significant effect of sex on jaw morphology (Pillai’s trace = 3.94e–30, F = 4.95e–29, p = 1) or of sex on the health assessment data (Pillai’s trace = 5.44e–31, F = 4.35e–30, p = 1). Taken together, there is little evidence for sexual dimorphism in this sample of pinfish.

The effect of ecomorphology on health assessment data

There is no evidence that ecomorphology (Table 3) or size (Table S2) represent the processes that generate the health assessment data. For the majority of blood traits that included sodium, potassium, calcium, glucose, the partial pressure of carbon dioxide, and saturated oxygen content, the best-fitting model was the null, intercept-only model (Table 3). While the model selection approach revealed some uncertainty in model choice with ΔAICc values less than 2 between null and second best model (Burnham & Anderson, 2002; Burnham & Anderson, 2004), quantification of the model-averaged estimates for all predictors across all models in the set ubiquitously yielded no evidence for a relationship between ecomorphology and hematology (Table 3).

Discussion

The results of our health assessment correspond with published work and provide new baseline information on additional blood parameters that can be used for monitoring the health status of this ecologically important species. We find no evidence for sexual dimorphism in either the morphological or hematological and biochemical datasets, consistent with previous studies that have found no sexual dimorphism in this species (Nelson, 2002). Integrating our health assessment with the morphological analyses we also find no evidence for a strong relationship between blood physiology and feeding morphology. Comparing our results to published values from other fish species supports hypotheses concerning patterns of hematocrit, lactate, and glucose content across marine fishes.

Towards developing baseline health statistics for wild pinfishes

Our analyses found no evidence for a relationship between blood parameters and either sex or quantified aspects of feeding ecology. Sexually dimorphic aspects of blood physiology have been documented across a diversity of vertebrates spanning fishes to mammals. For example, sexual dimorphism in hemoglobin and red blood cell counts have been found in the New World monkey Cebus apella (Rosner et al., 1985; N = 40) while hematocrit was found to vary significantly between male and female individuals of the shad Tenualosa ilisha (Jawad, Al-Mukhtar & Ahmed, 2004; N = 400). However, our results yielded no support for a relationship between any blood parameter and sex (Fig. 3 and Fig. S1). Given the overlap between sexes of each measured parameter (Fig. 3 and Fig. S1), this is unlikely a reflection of low sample size. Instead our inability to detect a relationship between sex and blood physiology in our study may be expected. Changes in hematology have been found to correspond with expected changes in the demands of hematological functions at different body masses (Burggren, Dupré & Wood, 1987) and sexual size dimorphism has been demonstrated for both Cebus apella (Masterson & Hartwig, 1998; Kay et al., 1988) and Tenualosa ilisha (Jafri et al., 1999). In contrast to these taxa, pinfish are not known to exhibit any degree of sexual size dimorphism (Nelson, 2002), a finding consistent with our morphological analyses (Fig. 3) even when accounting for potential variation due to patterns of allometry (Tables S1 and S2). Our findings are in line with previous blood work on this species that found no evidence for a relationship between sex and blood characteristics (Cameron, 1970). However, there are several reasons to expect that ecomorphology associated with habitat specificity could underlie hematological and biochemical differences.

Simple sand habitats are associated with more active roaming in pinfishes (Ruehl, Shervette & Dewitt, 2011), and a previous laboratory experiment linked increased activity with increased blood concentrations in this species (Cameron, 1970). This expectation of increased locomotor activity in sand habitats raises the question of whether sand habitat fidelity would promote detectable differences in blood physiology. As low sample sizes will bias towards simpler model fits given the penalty factor in a AICc framework (Burnham & Anderson, 2002), our inability to provide evidence for a link between ecomorphology and blood physiology could be an artifact of experimental design. Although our sample sizes are on par with other studies of fishes (e.g., Fazio et al.; n = 25), this is certainly plausible. However, we find sample size an unlikely explanation given the broad overlap of blood parameters and ecomorphology (Fig. S2). Alternatively, potentially confounding the expectation of habitat-specific differentiation of blood parameters is the observation that in more complex eel-grass dominated patches pinfishes focus foraging and feeding across a wider range of the water column, requiring continual fine-scale maneuvering (Ruehl, Shervette & Dewitt, 2011). In the absence of studies investigating the energetics of pinfishes in different habitat types, it is unclear how different these exercise regimes between habitats really are. Further, a not mutually exclusive hypothesis is that the mosaic nature of our collection site facilitates pinfish to opportunistically forage in both sandy and eel-grass dominated patches.

During our assessment multiple individuals representing the extremes of the measured variation were collected at the same exact (sand bottom) site, providing some evidence of migration between habitat patches. While juvenile pinfish show high site fidelity (Potthoff & Allen, 2003), movement patterns of adults are not known. Although a detailed study of pinfish activity patterns is outside the scope of our study, comparisons of our data with future data collected at habitat homogenous sites will offer a way to disentangle the possible effect of heterogeneous foraging patterns on baseline hematological and biochemical data. Likewise, integrating our study with a further investigation of juvenile fishes offers another unexplored axis of potential variation, as there was too little extractable blood within juveniles below 71 mm for the instruments used in this study. This forced us to restrict our sampling to adults, many of which are considered large for the species (Muncy, 1984). Regardless of possible mechanisms underlying the patterns of pinfish ecophysiology in our data, our findings contribute new baseline data and provide new axes of inquiry in future investigations of sources that underlie variation in blood physiology across the range of this important Atlantic fish.

Pinfish in the perspective of marine fish physiology

Ray-finned fishes (Actinopterygii) comprise over half of all living vertebrate species (Near et al., 2012b; Eschmeyer & Fricke, 2015), with over 17,000 species found in marine waters (Vega & Wiens, 2012). Although collecting baseline data across a quarter of living vertebrates is a task that will take several decades, existing work within pinfishes (Cameron, 1970) as well as several recent investigations of marine teleosts (Fazio et al., 2012; Francesco et al., 2012; Fazio et al., 2013) provide an early opportunity to compare other published blood parameters to values from our study.

Table 4 Descriptive statistics for hematology parameters from four species of marine fish as compared to pinfish (Lagodon rhomboides).

Values indicate mean and standard deviations for blood parameters. Cameron (1970) utilized heparinized capillaries and a scaled reading device to measure hematocrit and a cyanmethemoglobin method to measure hemoglobin. Fazio et al. (2013) utilized a HeCo vet C blood cell counter (SEAC) to quantify hematocrit and hemoglobin as well portable blood glucose (ACCU-Chek Active; Roche Diagnostics GmbH) and lactate (Accusport; Boehringer) analyzers for the additional parameters.

Hematological parameters	Lagodon rhomboides—1970 (N = not reported)	Gobius niger (N = 25)	Mugil cephalus (N = 25)	Sparus aurata (N = 25)	Dicentrarchus labrax (N = 25)	Lagodon rhomboides—this study (N = 37)	
Hematocrit (%)	32.1	22.47 ± 3.54	41.0 ± 3.31	53.33 ± 4.42	49.29 ± 6.17	34.39 ± 9.54	
Hemoglobin (g/dl)	7.6	5.67 ± 0.89	11.07 ± 1.05	9.95 ± 1.06	8.90 ± 0.76	11.7 ± 3.24	
Glucose (mg/dl)		168.90 ± 35.39	50.40 ± 8.40	192.8 ± 47.00	246.50 ± 30.93	169.7 ± 101.43	
Lactate (mmol/l)		3.38 ± 0.62	8.84 ± 1.99	6.36 ± 1.60	16.42 ± 1.68	9.93 ± 4.56	

Intraspecifically, our hematocrit values closely mirror previously collected data (Cameron, 1970; Table 4). Hematocrit values are closely related to blood oxygen content (Randall, 1982), suggesting similar arterial blood composition between populations. Our differential white blood cell count (WBC) also closely matches previous work by Cameron (1970), who found a wide range of variation, from 9.41 to 47.36 × 103/µl. Our pinfish WBC counts were similar in that the average total WBC count was 25.78 × 103/µl, ranging from 6.8 to 73.2 × 103/µl. As the defensive cells of the body, WBC levels have implications for immune response and the ability of the animal to fight infection, with higher levels correlated with more effective immune responses (Douglass & Jane, 2010). These wide ranges of WBC across two independent studies are enigmatic. While leucocrit is related to stress tolerances of individuals, such as physiological stress due to handling and temperature changes, WBC’s are relatively insensitive to physiological sampling procedures (Wedemeyer, Gould & Yasutake, 1983). This wide range could be attributed to immune function, heterogeneous environmental stressors, or even possibly infection or disease. More work is clearly needed to determine why pinfishes exhibit a WBC range spanning nearly an order of magnitude.

When comparing our pinfish blood values to values published for four other marine teleost fish species (Fazio et al., 2013), the hematocrit, hemoglobin, and lactate values most closely matched those of the flathead grey mullet (Mugil cephalus; Table 4). Fazio et al. (2013) also found that blood lactate concentrations were higher in more active fish, such as European sea bass (Dicentrarchus labrax) and mullet, compared to less active species (Table 4). The pinfish mean lactate value from our study was comparable to that of an active fish (Table 4), however glucose levels for this species differed substantially from both the European sea bass and mullet assessed by Fazio et al. (2013). While glucose measurements can vary greatly between analytical method and instrument, this large difference may reflect increasing carnivory impairing the ability to clear excess blood glucose levels (Cowey et al., 1977). Many carnivorous fish have been traditionally considered relatively glucose-intolerant species (Wilson, 1994; Moon, 2001), a hypothesis in line with the results of Fazio et al. (2013) who found the highest levels of glucose in the carnivorous European seabass and lowest in the herbivorous mullet. Since pinfish are omnivores (Montgomery & Targett, 1992), their glucose levels fall in the middle of this range, with values being closer to those of the omnivorous Gobius niger than the herbivorous M. cephalus.

Although taxonomic sampling is currently limited in marine fish hematological and biochemical studies, a comparison of our study with the results of Fazio et al. (2013) suggests that broad “ecohematological” patterns may well exist across the ray finned fish Tree of Life. Since the success of fishes is linked to their ability to diversify within any aquatic habitat type (Near et al., 2013), continual case studies of individual species will facilitate future investigations of correlations between blood value variation and ecology, and ultimately provide the necessary data to place fish blood physiology into a phylogenetic perspective. Such a historic perspective will not only help characterize the severity of physiological changes following different stress conditions, such as exposure to pollutants, disease, metals, hypoxia, and other stressors (Blaxhall, 1972; Duthie & Tort, 1985), but shed light on patterns of convergence in the physiological changes that underlie the successful diversification of fishes. As we move further in the 21st century, there is a growing consensus that marine fish species are valuable indicators from which to assess changes in ocean conditions in response to a variety of factors (Schlacher et al., 2007). Continual health assessments both within and between species will be pivotal to monitoring the health of the world’s oceans and offer the opportunity to gleen new insights into physiological processes that govern the origin and maintenance of marine biodiversity.

Supplemental Information

Data S1 Raw data form the pinfish health assessment and morphometrics

Click here for additional data file.

Figure S1 Box plot quartiles comparing female to male pinfish

Box plots comparing the quartiles of measured blood parameters between male and female pinfish.

Click here for additional data file.

Figure S2 Blood parameter to jaw angle scatterplots

Scatterplots comparing the distribution of hematological and blood chemistry values (log scale) to jaw angle measurements.

Click here for additional data file.

Table S1 Effect of body size on jaw morphology

The effect of body size on jaw morphology values. Coefficients and 95% confidence values for the SMA regression model testing for a possible effect of body size on each trait or significantly different effects between males and females.

Click here for additional data file.

Table S2 Effect of body size on hematological values

The effect of body size on hematological values. Coefficients and 95% confidence values for the SMA regression model testing for a possible effect of body size on each trait or significantly different effects between males and females. PCV, Packed cell volume; WBC, White blood cell count.

Click here for additional data file.

We would like to thank A Lamb and Dr. K Thompson for help catching pinfish in the field. We would also like to thank Dr. K Thompson, Dr. B Phillips, and Dr. C Harms for all of their help with the lab work. Dr. D Warren and Dr. T Iglesias provided valuable help and code for the general linear model analyses and R Morris provided additional help with other statistical analyses. Dr. A Camus assisted with deciphering blood samples and offering lab space in which to do hematological counts. We thank G Hogue, L Roupe, and L Lukas for help with specimen curation. We would like to thank K Passingham for logistical support and equipping us with the tools needed for the lab work.

Additional Information and Declarations

Competing Interests

Author Contributions

Animal Ethics

Data Availability

The authors declare there are no competing interests.

Sara Collins performed the experiments, analyzed the data, wrote the paper, prepared figures and/or tables, reviewed drafts of the paper.

Alex Dornburg conceived and designed the experiments, performed the experiments, analyzed the data, contributed reagents/materials/analysis tools, wrote the paper, prepared figures and/or tables, reviewed drafts of the paper.

Joseph M. Flores performed the experiments, reviewed drafts of the paper.

Daniel S. Dombrowski conceived and designed the experiments, conceived the initial idea and made the connections between the lead investigators.

Gregory A. Lewbart conceived and designed the experiments, contributed reagents/materials/analysis tools, reviewed drafts of the paper.

The following information was supplied relating to ethical approvals (i.e., approving body and any reference numbers):

This study was conducted at the North Carolina State University Center for Marine Sciences and Technology (CMAST) in Morehead City, North Carolina and approved by the IACUC ethics and animal handling protocol (15-005-O). All handling and sampling procedures were consistent with standard vertebrate protocols and veterinary practices.

The following information was supplied regarding data availability:

The raw data has been supplied as Supplemental File and at Zenodo DOI 10.5281/zenodo.58022.

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
