# Peer review of "A comparison of blood gases, biochemistry, and hematology to ecomorphology in a health assessment of pinfish (Lagodon rhomboides)"

_PeerJ, doi:10.7717/peerj.2262_

## Round 0.1 · original submission · Major Revisions

· Academic Editor

Major Revisions

Both reviewers have supplied good feedback - please incorporate and respond to their comments in any revision.

Please try to avoid vague words: regulate, modulate, affect, manipulate, interact, play role, target, closely related, dependent, mediated, impact. Instead use concrete words: increase, decrease, stimulate/activate, inhibit/suppress, bind, correlate.

·

Basic reporting

This article is well written, if a little wordy for the content. The data have been made available per journal policy. Some minor edits:

L76 check "the base of prey base for".
L89 I think you may mean "ontogeny".
L148 I think "the first" is meant to be "the fish".
L289 "Why then was?" is not a sentence.
L365 "gleam" should be "glean"

Experimental design

The knowledge gap being investigated (clinical pathology baselines in forage fish) is clearly identified and the sampling design was appropriate for the questions at hand: what are the impacts of gender and ecomorphotype on blood chemistry and haematology values in pinfish?

I'm glad to see the authors correcting for the effects of temperature on the i-Stat, but this can be done on the unit ahead of time, if memory serves me right. Check the setup menu.

Validity of the findings

I am concerned that the number of comparisons made is done in an uncorrected fashion. When so many pairwise tests of significant difference are made, a certain portion are bound to produce significance values below the chosen alpha. The Bonferroni correction exists to correct for the effects of multiple comparisons by scaling the alpha according to the number of comparisons. I think the authors may find that they have even fewer correlations than they found, if they corrected for multiple comparisons. I'm not sure how to correct in the case of AIC and recommend that the authors consult a statistician

It is a fortunate thing that PeerJ accepts negative results, because the essence of the paper is that gender and ecomorphotype were not, on the whole, correlated with any particular suite of blood changes. In a journal forum that did NOT accept negative results, I would be inclined to ask the authors to remove those figure panels and table lines that were not significant and leave only those that were (after Bonferroni correction of course!).

Comments for the author

As a paper simply presenting reference haematology and clinical chemistry values for pinfish, this manuscript would be fine, but would need to be cut down a lot. As an exploration of the relationships of ecology and blood values (i.e. as presented) it's somewhat unsatisfying because the results were generally negative, and because some of those that were significant might not be after correction for multiple comparisons. The paper reads like a student project that didn't produce the expected results or where the results were genuinely not suspected ahead of time. That's fine, and its OK to want to show evidence of all that work, but what does it add? If you remove the figures with non-significant r2 values and remove the lines in tables where p>0.05, what remains?

Reviewer 2 ·

Basic reporting

Overall, the manuscript is well written, with areas needing minor clarification and are noted by line number below. The Abstract and Introduction or Materials and Methods should include the number of fish in the experiment. The basic reporting criteria are met however the raw data on the Excel spreadsheet only shows WBC and differential results for 6 fish while the sample size in Table 2 shows n = 30.

Experimental design

I selected the category of Major Revision primarily due to the study design of using a small number of blood sample results to make the conclusions with regard to ecomorphology. This study was based on 37 fish. Per the tables of results, a total of 6 analytes had n=37 and the other 12 tests had fewer (n = 30-36). Also the number of males and females in this group of 37 is not given in the text or in the tables (found only in the supplemental material) therefore the comparisons by gender cannot be readily interpreted by the reader.

While the challenges of obtaining samples from free-ranging animals is certainly understandable, guidelines for calculating reference intervals for small sample sets are posted on the America Society for Veterinary Clinical Pathology website (www.asvcp.org) and published in Veterinary Clinical Pathology, see Vet Clin Pathol 41/4 (2012) 441-453. Another article that offers guidance for small sample sets is Geffre et al. Estimation of reference intervals from small samples: an example using canine plasma creatinine. Vet Clin Pathol 38/4 (2009) 477-484.

Lactate was measured using two different devices, the iSTAT and the lactate analyzer by Nova Biomedical. Did the two devices use different test methods? Only one set of data was reported. Which device was represented? Did you compare the results of the two devices?

WBC counts and differentials were performed using the 40x objective on the microscope. The similarly sized thrombocytes and lymphocytes found in non-mammalian species would be difficult to distinguish without using a 100x oil immersion lens. It would be helpful to include a figure of images to show the appearance of the cells at the 40x lens view.

Lines 228-232 explain that some of the samples were outside the analytical range of the iSTAT so the upper limit was used as the test result. This assumption influences the statistical analysis and the reader is unable to readily determine how many samples out of range (found in the supplemental materials).

Validity of the findings

Have there been studies in other species that show differences in hematology and blood gas values associated with allometric measurements? For example, would you expect the WBC count to be elevated? How would you distinguish this from changes associated with a stress leukogram?

Comments and Questions by Line
Abstract and Introduction: include the number of fish used in the study and the reason for selecting the sample size.
Line 112: “Fight times were minimized…” Do you mean minimal? Or was there an angling practice defined to minimize fight time?
Line 113-115: What was the approximate time of transport from site of collection to the laboratory? Was the laboratory seawater the same composition as the collection site? Was the water chemistry tested to determine if there were temperature or pH differences? Where were the fish housed when moved from the holding bucket to the laboratory tank? Or were they kept in the bucket? An acclimation period of “at least 5 minutes” is difficult to interpret. Were some fish allowed a longer rest period than others? Would you expect that capture, transport and acclimation stress could influence blood values more so than the body measurements?
Line 118: What blood volume was collected from the fish? (found in the supplemental table--it would be helpful to state in the text as well). How were the samples processed? What was the typical time interval between collection, the iSTAT, the Nova and the blood smear preparation?
Line 120: The iSTAT analyzer is no longer a Heska product. It is now owned by Abaxis.
Line 121: The CG8+ includes lactate. Please include the reason for running a second lactate test using a different analyzer. If a test comparison was performed, it was not reported in the manuscript. Or did you run some of the tests on one analyzer and some other on the other? If so, this is not noted in the text or tables.
Line 136: The manufacturer information for the centrifuge model is missing.
Lines 139-144: Were images recorded of the cells using the 40x objective? A figure showing the different cell types at that magnification would be helpful.
Line 146: In addition to photographs and dissection, were the fish weighed? (found in the supplemental table--it would be helpful to include in the published article)
Line 199 and throughout: the term hematological usually refers to the hematology tests (CBC) so I suggest “hematological and biochemical” dataset or similar.

Results—the size and weight of the fish is not reported. How big are the pinfish?

Line 230: Please indicate which tests and the number of samples that were out of analyzer range.
Line 270: “The results…greatly expand the number of hematological parameters that can be used” is a strong statement given the sample numbers of 30-37.
Line 272-3: Can you make this judgement with the number of males and females in this sample set? How many males/females were in this group?
Line 281: Are the studies cited here for other fish species? How many animals were used in the cited studies? Were the numbers similar to this study of 37 fish? The phrase “select parameters” is vague. Which parameters differed in the other animals?
Line 289: partial sentence “Why the was?”
Line 307: “increased activity has been linked to hematological parameters…” Which parameters have been reported to change? Do the blood samples collected reflect the activity level of active roaming or would you expect the influence of capture/transport/acclimation/sedation to also influence the blood test results?
Line 329-340: What was the sample size in the Cameron study (1970)? Is it possible that the wide range of WBC counts could be due to small sample size (<120) and/or the analytical technique? Did the Cameron study use a hemocytometer cell count or did they perform estimated counts? If an estimate, was the same technique used as in this study?
Line 346: It should be acknowledged that glucose measurement can vary greatly depending on analytical method and instrument, and this is not always known when comparing to previously published literature.
Figures: There are no units on Figures 3 and 4, or the font is so small that I can’t tell.
Table 1. With a small data set (<40), including the median is recommended. Adding a legend or notation to the table to indicate the number of the total n that exceed the analyzer range would be helpful. Is the lactate data from the iSTAT or the Nova analyzer? The title should state “minimum and maximum” rather than “range”. Range is the statistical value of the maximum less the minimum.
Table 2. The same comment for the title with respect to “range”. The WBC count should have units per uL.
Tables 3-5: Consult author guidelines for what abbreviations are acceptable and which need to be shown as full name in the legend.
Table 6: It would be very helpful if you would add the sample number to the table that was used in each referenced study. Add a legend that indicates the test methods used in the other papers. Suggested change for the title “Descriptive statistics for hematology parameters from four species of marine fish as compared to the pinfish (Lagodon rhomboides)” In the table, use scientific names for the pinfish as for the other species.

---

## Round 0.2 · Minor Revisions

· Academic Editor

Minor Revisions

Please address the comment from reviewer.

Reviewer 3 ·

Basic reporting

The basic reporting is written in a very simple and lucid manner. Although I feel the introduction seems a bit lengthy. The authors might want to cut it down.

Experimental design

The experimental design for the study only includes ~37 fish. A correlation study with such small dataset might be biased. Further comparison based on the gender makes the dataset more smaller.

Validity of the findings

I find the the findings very interesting.

Comments for the author

The authors have answered all the concerns of the previous reviewers accurately and to the point. I have only one concern with the paper. The experimental design for the study only includes ~37 fish. A correlation study with such small dataset might be biased. Further comparison based on the gender makes the dataset more smaller.

---

## Round 0.3 · accepted · Accept

· Academic Editor

Accept

Please ensure your manuscript format is properly formatted to PeerJ style.